# Skin Microbiota, Immune Cell, and Skin Fibrosis: A Comprehensive Mendelian Randomization Study

**DOI:** 10.3390/biomedicines12102409

**Published:** 2024-10-21

**Authors:** Zirui Zhao, Yanchao Rong, Rong Yin, Ruixi Zeng, Zhongye Xu, Dongming Lv, Zhicheng Hu, Xiaoling Cao, Bing Tang

**Affiliations:** 1Department of Burns, Wound Repair and Reconstruction, First Affiliated Hospital of Sun Yat-sen University, Guangzhou 510080, China; zhaozr5@mail2.sysu.edu.cn (Z.Z.); rongyanchao_1995@163.com (Y.R.); xuzhye@foxmail.com (Z.X.); lvdongm@126.com (D.L.); willway128@163.com (Z.H.); 2Department of Dermatology, First Affiliated Hospital of Sun Yat-sen University, Guangzhou 510080, China; yinr9@mail.sysu.edu.cn; 3Department of Plastic Surgery, First Affiliated Hospital of Sun Yat-sen University, Guangzhou 510080, China; zengrx@mail.sysu.edu.cn

**Keywords:** skin microbiota, leaky epithelia, skin fibrosis, immune cell, mediation analysis

## Abstract

Background: Microbiota dysbiosis has been reported to lead to leaky epithelia and trigger numerous dermatological conditions. However, potential causal associations between skin microbiota and skin fibrosis and whether immune cells act as mediators remain unclear. Methods: Summary statistics of skin microbiota, immune cells, and skin fibrosis were identified from large-scale genome-wide association studies summary data. Bidirectional Mendelian randomization was performed to ascertain unidirectional causal effects between skin microbiota, immune cells, and skin fibrosis. We performed a mediation analysis to identify the role of immune cells in the pathway from skin microbiota to skin fibrosis. Results: Three specific skin microbiotas were positively associated with skin fibrosis, while the other three were negative. A total of 15 immune cell traits were associated with increased skin fibrosis risk, while 27 were associated with a decreased risk. Moreover, two immune cell traits were identified as mediating factors. Conclusions: Causal associations were identified between skin microbiota, immune cells, and skin fibrosis. There is evidence that immune cells exert mediating effects on skin microbiota in skin fibrosis. In addition, some strains exhibit different effects on skin fibrosis in distinct environments.

## 1. Introduction

Senescence is a complex physiological phenomenon occurring in various human tissues and organs. Skin senescence has attracted considerable attention owing to its effects on the largest organ in the human body [1]. Skin fibrosis, a chronic dermatological condition characterized by a disruption of skin homeostasis, can be considered a process of skin senescence. Skin fibrosis includes hypertrophic scarring and localized scleroderma [2]. The damage to the skin may result in impairment of the skin barrier and disruption of skin homeostasis, which can lead to leaky epithelia and disease [3,4]. Furthermore, aberrant secretion of the senescence-associated secretory phenotype (SASP) can trigger immune responses and foster fibrosis in the skin [5]. Improving our understanding of skin fibrosis is expected to improve patients’ quality of life and provide insights into novel targets for addressing skin senescence.

The etiology of dermatological conditions is multifactorial, and the role of bacterial microorganisms has become increasingly recognized in recent years [6]. Hidradenitis suppurativa, acne vulgaris, rosacea, alopecia areata, atopic dermatitis, and psoriasis have been proven to be associated with gut microbiota. *Bacteroides* were reported to maintain the integrity of the skin barrier [4]. Opportunistic bacteria may invade the dermis via the leaky epithelium, leading to lesions [7,8,9]. Moreover, gut microbiota has the potential to invade the bloodstream through leaky epithelia and reach various tissues in the human body, such as those along the gut–skin, gut–lung and gut–brain axes [6,7]. Furthermore, researchers have reported that gut microbiota plays an important role in skin senescence, as gut dysbiosis can promote the release of SASP, which leads to senescence [10,11]. SASP is also a significant contributor to the development of skin fibrosis [5]. Considering these findings, further investigation into the role of microbiota will significantly improve our understanding of both skin fibrosis and senescence. Our previous research showed that gut microbiota may influence skin homeostasis and contribute to skin fibrosis [12]. Compared with gut microbiota, skin microbiota may be more variable [13]. However, there is a lack of research examining the role of skin microbiota in skin fibrosis. This article aimed to explore the impact of skin microbiota on fibrotic conditions such as hypertrophic scarring and localized scleroderma.

The immune system plays a pivotal role in regulating the host interactions with the gut microbiota [7]. Microbiota was reported to affect immune responses by promoting macromolecule and antigen transport through the epithelium. The flagellin of microbiota is recognized by TLR5 on B cells, which differentiate into cells capable of producing IgA to neutralize pathogens and prevent infection [6,14]. *S. aureus* α-toxin can induce IL-1β production from monocytes, thus activating an immune response. [15]. SASP is an inflammatory mediator that can also induce immune responses [1,5]. Microbial dysbiosis has been reported to promote SASP damage [11]. Microbiota can release proinflammatory microbial products into the bloodstream via leaky epithelia, resulting in immune cross-talking immune [16].

The substantial advancement of genome-wide association studies (GWAS) has contributed significantly to clarifying the relationship between potential pathogenic factors and diseases [17]. As genetic variations are randomly assigned during meiosis and are independent of environmental and other acquired factors, a Mendelian randomization (MR) analysis can be performed to infer a credible causal relationship with single nucleotide polymorphisms (SNPs) of GWAS [18,19].

We performed a comprehensive MR analysis to elucidate associations between skin microbiota, immune cells, and skin fibrosis. A Mediation analysis was performed to identify whether immune cells mediated the effect of skin microbiota on skin fibrosis. Our findings may improve the treatment of skin fibrosis and prevention of skin senescence.

## 2. Materials and Methods

### 2.1. Study Design

The study design is summarized in the flowchart in Figure 1A. A two-step MR was performed to determine the effects of mediation. In step 1, the impact of skin microbiota on skin fibrosis was determined by bi-directional MR. In step 2, skin fibrosis-related immune cell responses were determined by bi-directional MR. Then, causal effects of skin fibrosis-related skin microbiota on immune cells were established. In step 3, a mediated analysis was performed to verify whether immune cells mediated the impact of skin microbiota on skin fibrosis (Figure 1B) [20,21].

### 2.2. Data Preparation

Genetic data for the skin microbiota (GCST90133164-GCST90133313) were extracted from the GWAS Catalog [22]. This dataset comprised 72 skin microbiota samples obtained from dry skin, 53 samples from moist skin, and 22 samples from sebaceous skin; three unknown taxa were excluded. The GWAS Catalog (GCST90001391-GCST90002121) was used to extract summary statistics of immune cells, yielding 731 immune cell traits [23]. The GWAS summary data of the Hypertrophic scar and Localized scleroderma traits were extracted from the 10th Finngen consortium [24]. Detailed information is provided in Appendix A.

### 2.3. SNPs Selection

SNPs associated with each trait (*p* < 1 × 10^−5^) were extracted for further analysis [25]. The red line represents the threshold (*p* < 1 × 10^−5^) in the Hypertrophic scar and Localized-scleroderma traits (Figure 2A,B). The linkage disequilibrium of SNPs met the condition with the EUR population reference (r^2^ < 0.01 and clump distance > 10,000 kb). SNPs with an *F*-statistic < 10 were excluded to avoid weak bias. Palindromic SNPs were removed after matching the outcome [26].

### 2.4. Bi-Directional MR Analysis

An MR and reverse MR analysis were performed to explore the causal effects of skin microbiota and immune cells on skin fibrosis, respectively (Figure 1B). We choose the inverse variance weighted (IVW) approach as the primary analytic method to ensure robust estimation [27]. Statistical significance was determined at *p*-value < 0.05.

### 2.5. Sensitivity Analysis

We removed outliers and corrected the horizontal plural effect by using MR-Pleiotropy Residual Sum and Outlier (MR-PRESSO) [28]. Then, a leave-one-out analysis was performed to validate the results [26]. Cochran’s Q test was performed for heterogeneity; the Q statistic *p*-value > 0.05 indicated no heterogeneity. The MR-Egger test was performed for horizontal pleiotropy; the *p*-value > 0.05 indicated no pleiotropy [29,30].

### 2.6. Mediation Analysis

After determining the impact of skin microbiota and immune cells on skin fibrosis, a two-step MR was performed to determine whether immune cells mediate the causal pathway between skin microbiota and skin fibrosis (Step 3 in Figure 1B). The impact of skin microbiota on skin fibrosis was identified as β (Total effect), that of skin microbiota on immune cells was identified as β_1_, and that of skin microbiota on skin fibrosis was identified as β_2_. The mediated effect was determined as (β_M_) = β − (β_1_ × β_2_) [31]. 

All analyses were performed by R (version 4.3.2) and TwoSampleMR package (version 0.5.10).

## 3. Results

### 3.1. Step 1, Causal Effects of the Skin Microbiota on Skin Fibrosis

A total of six skin microbiota were associated with skin fibrosis. *Enhydrobacter* (unc.) (moist skin) was positively associated with hypertrophic scar (odds ratio [OR] 1.060 [95% confidence interval [CI] 1.015, 1.108], *p* = 0.008). *Anaerococcus* (unc.) (dry skin) was negatively associated with hypertrophic scar (OR 0.954 [95% CI 0.941, 0.996], *p* = 0.031). Class: betaproteobacteria (moist skin) was associated with a decreased risk of hypertrophic scar (OR 0.938 [95% CI 0.894, 0.985], *p* = 0.010) (Figure 3A). *S. epidermidis* (moist skin) showed potentially negative associated with local scleroderma (OR 0.917 [95% CI 0.844, 0.996], *p* = 0.039), while *S. epidermidis* (dry skin) showed potentially positive relationship with local scleroderma (OR 1.145 [95% CI 1.043, 1.257], *p* = 0.004). Besides, *R. mucilaginosa* (dry skin) was positively associated with local scleroderma (OR 1.105 [95% CI 1.005, 1.215], *p* = 0.040) (Figure 3B). Sensitivity analyses showed no horizontal pleiotropy by MR-Egger test, no significant heterogeneity by Cochran’s Q test and no horizontal pleiotropy by an MR-PRESSO analysis. All *p*-values were >5 × 10^−2^. Detailed information was provided in Appendix A. There was no reverse effect between these skin microbiota and skin fibrosis (Appendix A). The reliability was proved by a leave-one-out analysis (Appendix A). Causal associations between skin microbiota and skin fibrosis were exhibited in forest plots (Appendix A). The overall effect of skin microbiota on skin fibrosis was shown by scatter plots (Appendix A). Funnel plots were also performed to show the heterogeneity of evaluated SNPs in skin microbiota (Appendix A).

### 3.2. Step 2, Causal Effects of the Immune Cell on Skin Fibrosis

42 immune cells showed potential associated with skin fibrosis (Figure 4A,B). The closer the colour is to red, the smaller the *p*-value; conversely, the closer it is to blue, the larger the *p*-value. CD62L^−^ Dendritic Cell, Immature Myeloid-Derived Suppressor Cells, HLA DR^+^ CD4^+^ T cell, BAFF-R on CD20^−^ CD38^−^ B cell, CD24 on IgD^+^ CD38+ B cell, CD38 on CD3^−^ CD19^−^, BAFF-R on CD20^−^ B cell, CD25 on CD45RA^−^ CD4 not regulatory T cell and CD45RA on CD39^+^ resting CD4 regulatory T cell were potentially positive associated with hypertrophic scar, while CD25^++^ CD45RA^−^ CD4 not regulatory T cell, Naive CD4^−^CD8^−^ T cell, CD8^+^ T cell, CD4^+^ CD8dim T cell, CD28^+^ CD45RA^−^ CD8dim T cell, CD3 on CD39^+^ resting CD4 regulatory T cell, HLA DR on CD14^+^ CD16^−^ monocyte, HLA DR on CD14^+^ monocyte, CD14 on CD33dim HLA DR^+^ CD11b^+^ Myeloid cell, CD39 on monocyte and CD45 on Immature Myeloid-Derived Suppressor Cells were potentially negative associated with hypertrophic scar. CD33dim HLA DR^+^ CD11b^+^ Myeloid cell, CD45RA^−^ CD28^−^ CD8^+^ T cell, CCR7 on naive CD8^+^ T cell, CD45 on CD14^+^ monocyte, PDL-1 on CD14^−^ CD16^+^ monocyte and SSC-A on lymphocyte were associated with an increased risk of local scleroderma, while IgD^−^ CD24^−^ B cell, IgD^−^ CD38dim B cell, CD25^++^ CD45RA^+^ CD4 not regulatory T cell, Terminally Differentiated CD4^+^ T cell, Natural Killer, CD25^++^ CD8^+^ T cell, CD3 on Naive CD4^+^ T cell, CD3 on Central Memory CD8^+^ T cell, HVEM on Effector Memory CD8^+^ T cell, CD127 on CD45RA^+^ CD4^+^ T cell, CCR2 on CD14^+^ CD16^+^ monocyte, CX3CR1 on CD14^−^ CD16^+^ monocyte, CD14 on Monocytic Myeloid-Derived Suppressor Cells, CD8 on Terminally Differentiated CD8^+^ T cell, CD45RA on naive CD4^+^ T cell and CD45RA on naive CD8^+^ T cell were associated with a decreased risk of local scleroderma. Detailed information was provided in Appendix A. There was no reverse effect between these immune cells and skin fibrosis (Appendix A).

### 3.3. Step 3, Mediation Analysis

A bi-directional MR analysis was performed for the relationship between skin microbiota and immune cells. *Enhydrobacter* (unc.) (moist skin) was negatively associated with CD14 on CD33dim HLA DR^+^ CD11b^+^ Myeloid cell (OR = 0.950, 95% CI = 0.907–0.994, *p* = 0.028) (Figure 5A). *S. epidermidis* (moist skin) was negatively associated with CD14 on Monocytic Myeloid-Derived Suppressor Cells (OR = 0.947, 95% CI = 0.907–0.989, *p* = 0.014). *R. mucilaginosa* (dry skin) was positively associated with CD33dim HLA DR^+^ CD11b^+^ Myeloid cell (OR = 1.040, 95% CI = 1.002–1.080, *p* = 0.041) (Figure 5B). Detailed information was provided in Appendix A. Moreover, CD14 on CD33dim HLA DR^+^ CD11b^+^ Myeloid cell was associated with a decreased risk of hypertrophic scar (OR = 0.910, 95% CI = 0.844–0.982, *p* = 0.015) (Figure 6A). CD33dim HLA DR^+^ CD11b^+^ Myeloid cell was associated with an increased risk of local scleroderma (OR = 1.161, 95% CI = 1.007–1.339, *p* = 0.040), and CD14 on Monocytic Myeloid-Derived Suppressor Cells was associated with a decreased risk (OR = 0.895, 95% CI = 0.803–0.997, *p* = 0.044) (Figure 6B).

The mediated effect was calculated by the product of the coefficients method (Appendix A). We found that *S. epidermidis* (moist skin) did not affect local scleroderma via CD14 on Monocytic Myeloid-Derived Suppressor Cells (*p* = 0.015). Frost plots were performed to summarize the association between skin microbiota, immune cells, and skin fibrosis (Figure 7A and Figure 8A). A mediated effect analysis showed evidence of the mediated effect of *Enhydrobacter* (unc.) (moist skin) on the *Hypertrophic scar* trait through CD14 on CD33dim HLA DR^+^ CD11b^+^ Myeloid cell, with a mediated proportion of 8.3% (95% CI = 1.22–15.4%, *p*  =  0.0215) of total effect (Figure 7B). We observed an indirect impact of *R. mucilaginosa* (dry skin) on local scleroderma through CD33dim HLA DR^+^ CD11b^+^ Myeloid cell, with a mediated proportion of 5.86% (95% CI = 0.207–11.5%, *p*  =  0.0422) of the total effect (Figure 8B).

## 4. Discussion

Our findings indicated a causal association between skin microbiota and skin fibrosis. The presence of three specific types of skin microbiota (*Enhydrobacter* (unc.) (moist skin), *S. epidermidis* (dry skin), and *R. mucilaginosa* (dry skin)) were associated with an increased risk of skin fibrosis, while *Anaerococcus* (unc.) (dry skin), Class: betaproteobacteria (moist skin), and *S. epidermidis* (moist skin) were associated with a decreased risk. Furthermore, two immune cell traits (CD33dim HLA DR^+^ CD11b^+^ Myeloid cell and CD14 on CD33dim HLA DR^+^ CD11b^+^ Myeloid cell) exhibited mediating functions on the effect of skin microbiota on skin fibrosis.

The skin is an epithelial barrier to the external environment. The destruction of the skin barrier function is an essential component in numerous skin diseases [4]. The hypersaline and acidic environment of the skin barrier, coupled with low nutrient availability, distinguishes it from other mucosa and epithelia [32]. Diverse microbial communities have been associated with the skin barrier [33]. Microbial dysbiosis can potentially disrupt cutaneous homeostasis, resulting in leaky epithelia and disease risk [3,4]. A comprehensive investigation of pathogenic microorganisms can help guide strategies for maintaining cutaneous homeostasis and preventing leaky epithelia. Next-generation sequencing technologies in microbiological identification have helped us improve our understanding of the microbiota [13]. However, most metagenomic cataloging of the human microbiome has concentrated on species composition. The difference in density and variety of glands and hair follicles leads to the diversity of the skin environment [34]. The same strain can colonize multiple body parts and exhibit distinct characteristics according to the colonization site [35]. Our findings indicated that *S. epidermidis* on moist skin was associated with a reduced risk of local scleroderma, while *S. epidermidis* on dry skin exhibited the opposite. *S. epidermidis* is frequently found in moist areas of the skin and is proven to play a role in maintaining the skin barrier by producing protective ceramides [36,37]. It has been reported that *Enhydrobacter* is present in higher relative abundance in older skin, especially in moist areas [38]. Compared with lesion skin, *Anaerococcus* mainly presents in healthy and non-lesion skin, indicating a relation to cutaneous homeostasis [39]. The presence of *R. mucilaginosa* on patients’ skin was more frequently observed than the healthy, indicating an increased risk of leaky epithelia and subsequent infection in patients with severe skin barrier disruption, such as patients with extensive burns [40]. A reduction in diversity and the proportion of Proteobacteria have also been observed in individuals with atopic dermatitis compared to the healthy [15,41]. Further research would undoubtedly prove beneficial in elucidating the role of microbiota in various disease processes.

Many immune cells engage in constant communication with the gut microbiota within the gastrointestinal tract. The maturation of the immune system requires the development of commensal microorganisms. Furthermore, the gut microbiota is capable of mediating neutrophil migration and influencing T-cell differentiation, which may induce an immune response and stimulate inflammation or chronic tissue damage [6]. Skin microbiota are likely to affect many immune-related properties of epithelial health that are not yet fully known [34]. Dysbiosis can disrupt cutaneous homeostasis, leading to leaky epithelia. This dysbiosis enables the penetration of microorganisms or their metabolites through the intercellular cracks in the skin, resulting in crosstalk between the microbiota and the immune system [3,42]. According to the mediation analysis, the myeloid cells subtype may mediate the impact of skin microbiota on skin fibrosis. It has been reported that the infiltration and maturation of myeloid cells play a role in fibrotic repair [43]. Myeloid cells are a broad category of immune cells derived from myeloid progenitors in the bone marrow, including neutrophils. Myeloid cells function as potent producers of proinflammatory or anti-inflammatory factors and actively contribute to the pathogenesis of inflammatory diseases [44]. Neutrophils play a pivotal role in tissue restoration during wound healing by limiting microbial invasion at sites of skin or mucosal injury [45]. These findings highlight a novel avenue for research into the treatment of skin fibrosis and postponement of senescence. Furthermore, a more profound understanding of the interactions between the microbiota and immune cells may facilitate the development of more efficacious treatments.

The present study has some limitations. Firstly, the identification and classification of several microbiota were based on amplicon variant sequences, which may not fully capture the complexity of microbial communities [46]. Second, it would be beneficial to ascertain whether these findings apply to other ethnic groups despite the extensive sample size, given that the study participants were of European ancestry, and although the sample size is large, it would be beneficial to avoid generalizations before ascertaining whether these findings apply to other ethnic groups [20]. Thirdly, this study employed large MR analyses on 147 skin microbiota traits, 731 immune cell traits, and two skin fibrosis traits; it is difficult to perform Bonferroni correction to obtain statistically significant results. Therefore, caution should be exercised when interpreting results with IVW-derived *p* values less than 0.05. However, bidirectional MR was performed to determine the unidirectional nature of the causal effects between skin microbiota, immune cells, and skin fibrosis. Sensitivity analyses showed that our results were reliable. The findings of the study were consistent with those of previous research. Further clinical trials will be helpful in corroborating the reliability of our findings and facilitating the development of more efficacious treatments.

## 5. Conclusions

The concept of leaky epithelia has emerged as a novel and critical factor in the development and progression of skin diseases, including fibrosis and senescence. Microbial dysbiosis plays a pivotal role in disrupting the natural cutaneous homeostasis, which weakens the epithelial barrier. This can lead to leaky epithelia, which is now considered a key contributor to various skin disorders. We report a casual association between skin microbiota and fibrosis, with immune cells acting as mediators. Skin microbiota has a pivotal function in skin fibrosis and can induce immune responses in the process, such as the impact of *R. mucilaginosa* (dry skin) on skin fibrosis through myeloid cell interactions. It is crucial to investigate the role of specific skin microbiota in dermatological conditions for precise and personalized treatments of different strains comprising senescence. For example, non-invasive skin swabs can be used to collect the skin microbiota for early screening and diagnosis. For patients diagnosed with skin fibrosis, we propose novel and effective therapeutic strategies targeting specific skin microbiota for restoring the balance of the skin microbiome and mitigating the immune response. The effects of treatment can be evaluated by detecting the relative abundances of skin microbiota.

In future research, single-cell technologies combined with genomic, transcriptomic, proteomic and metabolomic analyses will clarify the crosstalk between the microbiota and the immune system, thus promoting our understanding of the mechanism of skin fibrosis. These strategies can then guide the selection of novel therapeutic targets for skin senescence. Ultimately, this research will offer deeper insights into the microbial contributions to skin health and aging, which can support innovative strategies for combating skin fibrosis and promoting long-term skin vitality.

## Figures and Tables

**Figure 1 biomedicines-12-02409-f001:**
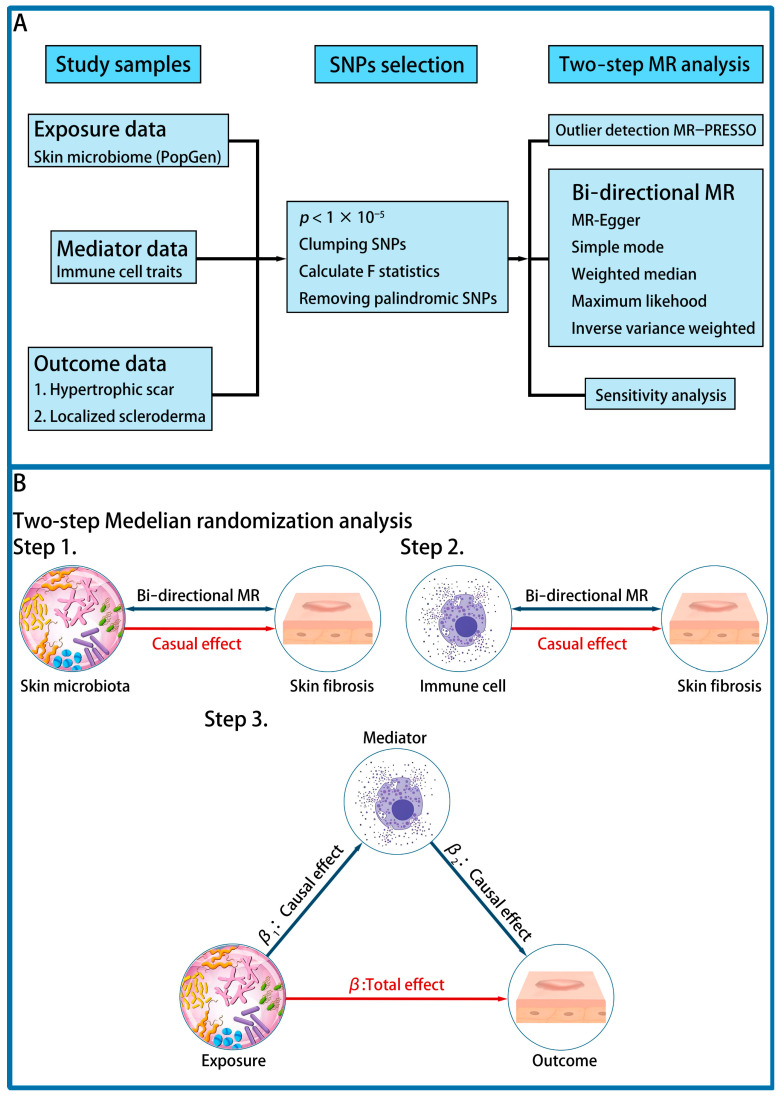
The design of the study. (**A**) The flowchart of the study. (**B**) Step 1, the causal effect of skin microbiota on skin fibrosis revealed by bi-directional MR. Step 2, The impact of immune cells on skin fibrosis revealed by bi-directional MR. Step 3, the mediated effect of skin microbiota on skin fibrosis through immune cells revealed by mediation analysis.

**Figure 2 biomedicines-12-02409-f002:**
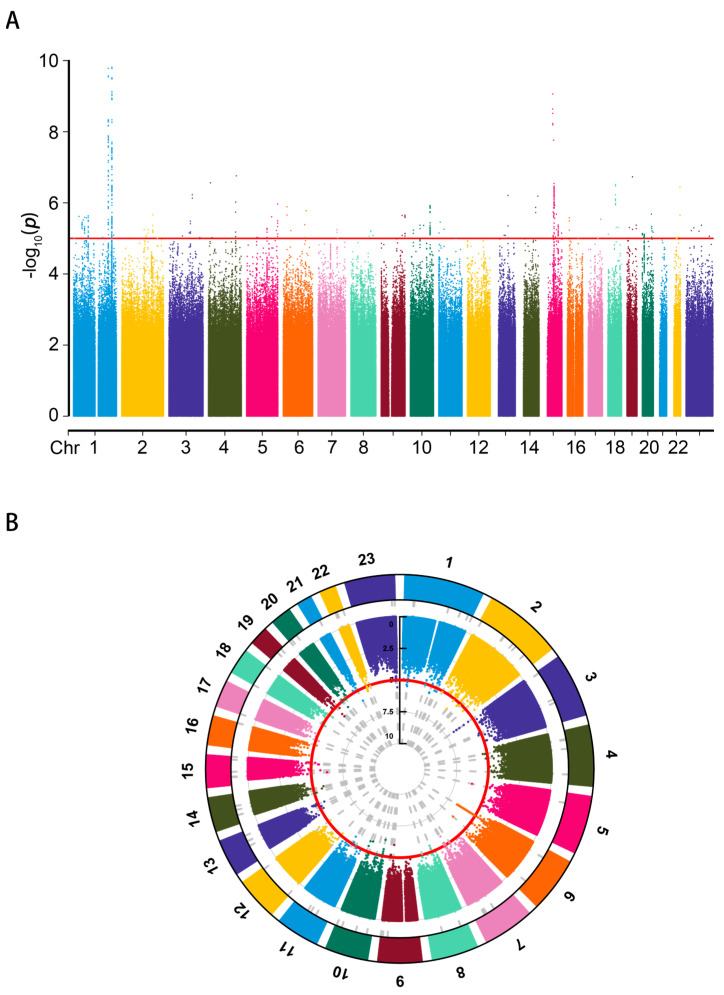
The threshold of SNP selection. (**A**) The Manhattan plot of the Hypertrophic scar trait (L12_HYPETROPHICSCAR) (threshold = 1 × 10^−5^). (**B**) The circle Manhattan plot of the Localized scleroderma trait (L12_LOCALSCLERODERMA) (threshold = 1 × 10^−5^).

**Figure 3 biomedicines-12-02409-f003:**
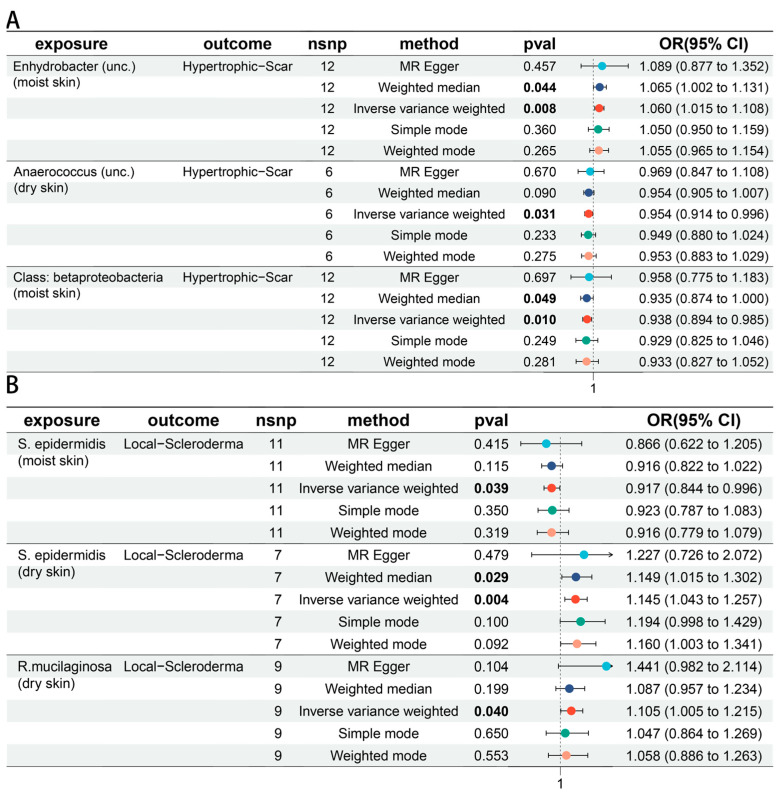
The impact of skin microbiota on skin fibrosis. (**A**) Forest plots of causal effects between skin microbiota and Hypertrophic scar trait. (**B**) Forest plots of causal effects between skin microbiota and Localized scleroderma trait.

**Figure 4 biomedicines-12-02409-f004:**
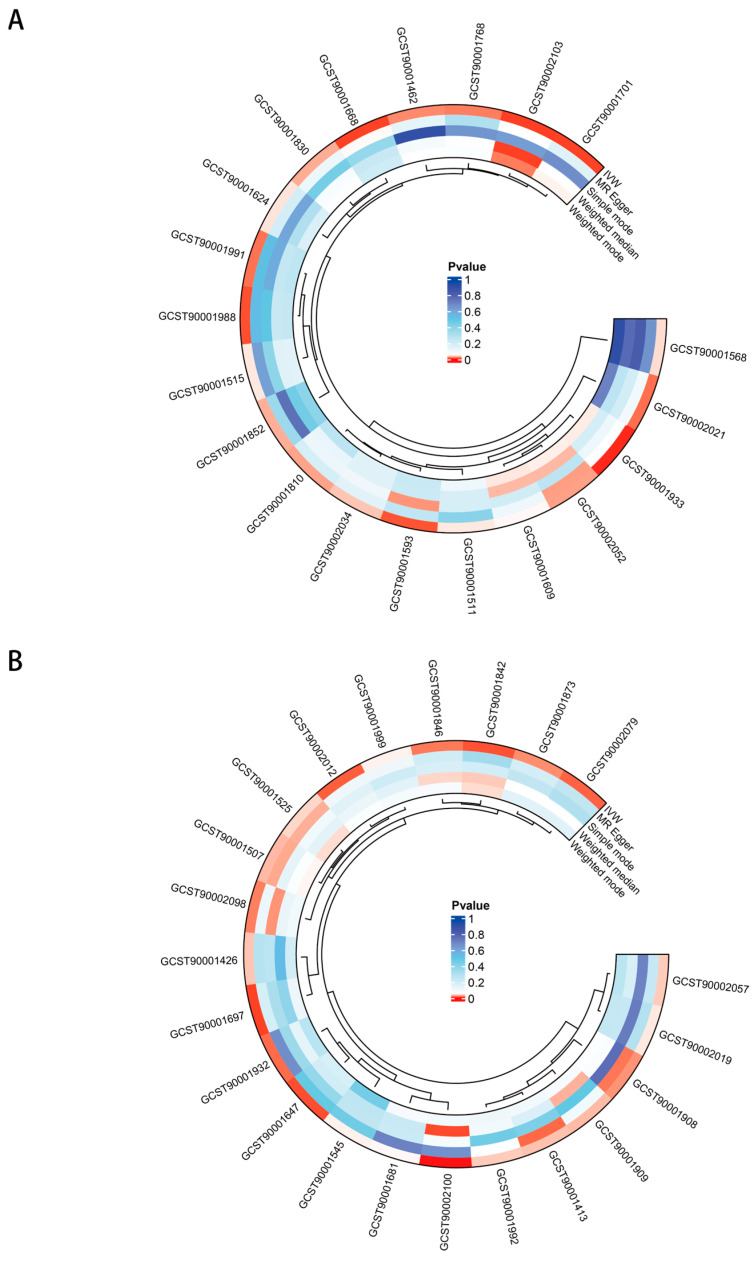
The impact of immune cells on skin fibrosis. (**A**) The circle plot of causal effects between immune cell and Hypertrophic scar trait. (**B**) The circle plot of causal effects between immune cells and Localized scleroderma trait.

**Figure 5 biomedicines-12-02409-f005:**
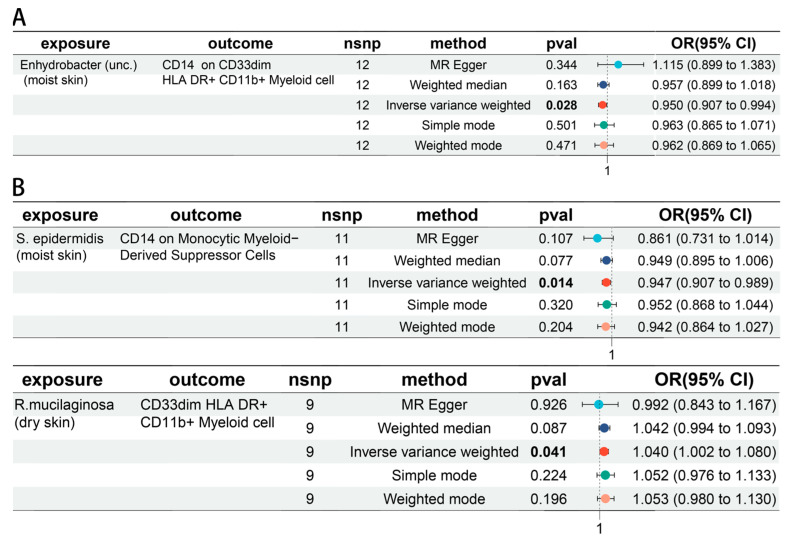
The causal association between skin microbiota and immune cells. (**A**) The forest plot of causal effects between Hypertrophic scar trait-related skin microbiota and immune cells. (**B**) Forest plots of causal effects between Localized scleroderma trait-related skin microbiota and immune cells.

**Figure 6 biomedicines-12-02409-f006:**
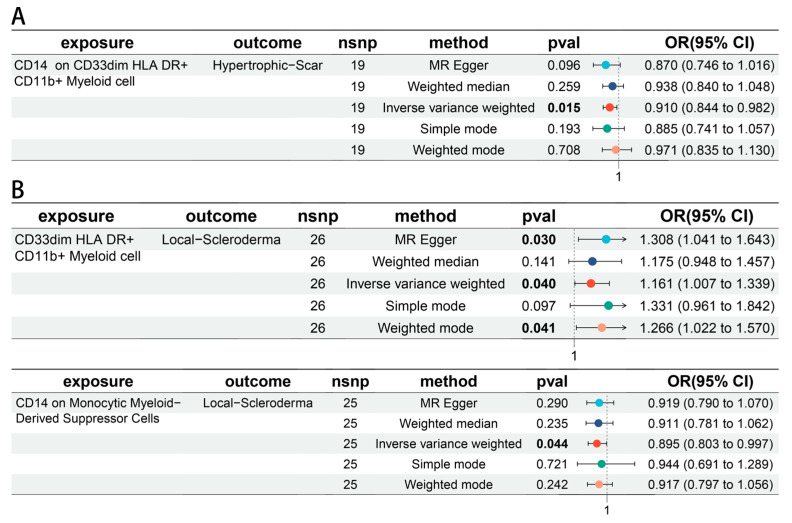
The causal association between immune cells and skin fibrosis. (**A**) The forest plot of a causal effect between immune cell and Hypertrophic scar trait. (**B**) Forest plots of causal effects between immune cell and Localized scleroderma trait.

**Figure 7 biomedicines-12-02409-f007:**
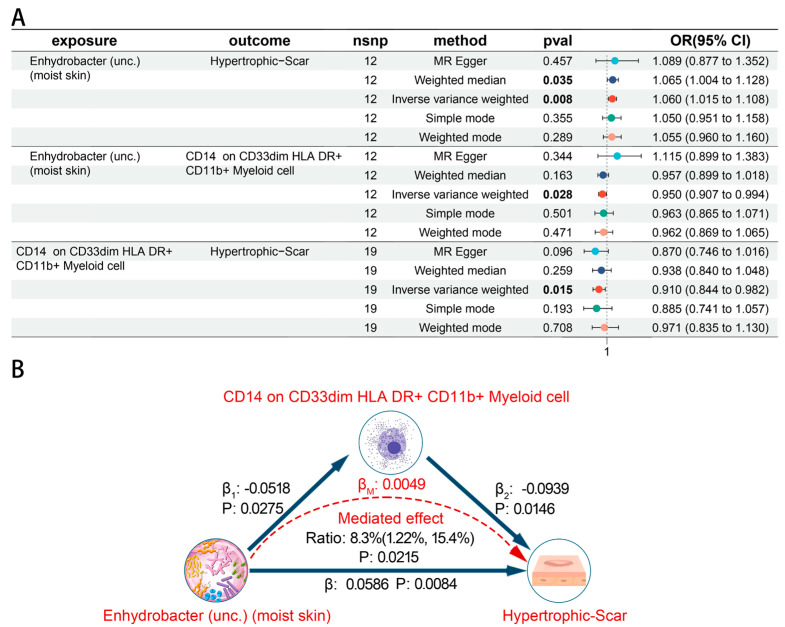
Causal effects between skin microbiota, immune cells, and Hypertrophic scar trait. (**A**) Forest plots of causal effects between skin microbiota, immune cells, and Hypertrophic scar trait. (**B**) The mediated effect of skin microbiota on Hypertrophic scar trait through immune cells.

**Figure 8 biomedicines-12-02409-f008:**
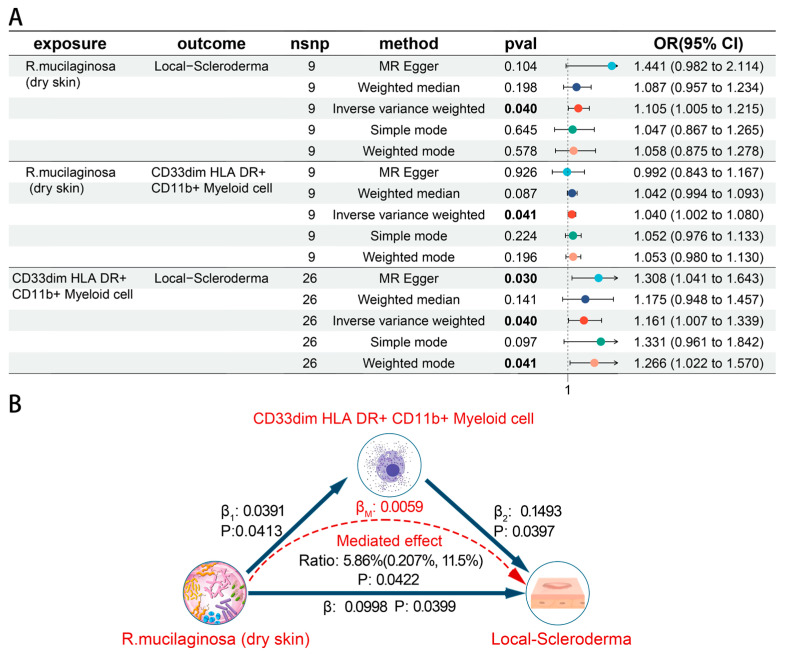
Causal effects between skin microbiota, immune cells, and Localized scleroderma trait. (**A**) Forest plots of causal effects between skin microbiota, immune cells, and Localized-scleroderma trait. (**B**) The mediated effect of skin microbiota on Localized scleroderma trait through immune cells.

## Data Availability

The data used to support the results are available at GWAS Catalog (www.ebi.ac.uk/gwas/home), and FINNGEN (https://r10.risteys.finngen.fi/). The accessed date was 6 June 2024 for both of them.

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
