# Peer review of "Skin Microbiota, Immune Cell, and Skin Fibrosis: A Comprehensive Mendelian Randomization Study"

_biomedicines, 2024, doi:10.3390/biomedicines12102409_

Round 1

Reviewer 1 Report

Comments and Suggestions for Authors

Zhao et al. reported on skin microbiota, immune cell function and skin fibrosis.  An interesting report, but it could be improved.

Minor:

Figures 2, 3, 4 and 5 the labels are too small to read, please correct.

Major:

In the discussion section:

1.        R.mucilaginosa (dry skin) was associated with an increased risk of skin fibrosis, but not covered?

2.        betaproteobacteria (moist skin) was associated with a decreased risk, but not covered?

3.        The function of the immune cell traits as mediators of skin fibrosis is not discussed

4.        How are the biomarkers above involved in mediating skin fibrosis (skin scarring), which is  pathological wound healing, where TGF-beta is released from macrophages that involve AKT/mTOR and SMAD pathways?

Conclusions:

This section is vague.  While this topic is important to study, what is the value to the readership in how this applies to skin fibrosis now and in the future?

Author Response

Dear Reviewer:

Many thanks to the peer expert for reading our manuscript and reviewing it, which will help us improve it to a better scientific level. We revised our manuscript, and quite a lot of changes have taken place. We have resubmitted the revised manuscript and changes to our manuscript are all highlighted within the document by using red coloured text. We hope that the new manuscript will be acceptable to you. The following, points mentioned by the reviewer will be discussed:

Response to Minor issue

Figures 2, 3, 4 and 5 the labels are too small to read, please correct.

Response

We appreciate your valuable comment and modified all figures. The changes of figure numbers and legends were highlighted within the document by using red colored text.

Response to Major issues

Q1. R.mucilaginosa (dry skin) was associated with an increased risk of skin fibrosis, but not covered?

Response

Thank you for your comment. In the submitted version, we added the sentence to cover R.mucilaginosa (dry skin) “The presence of R. mucilaginosa on patients’ skin was more frequently observed, indicating an increased risk of leaky epithelium leading to infection for severe patients, such as patients with extensive burns [1].” at section “4. Discussion”.

Q2. In Figure2. What does the dotted black line indicate betaproteobacteria (moist skin) was associated with a decreased risk, but not covered?

Response

Thank you for your careful review. “A reduction in diversity and the proportion of proteobacteria have also been observed in individuals with atopic dermatitis compared to the healthy [2,3].” was added in section “4. Discussion

Q3. The function of the immune cell traits as mediators of skin fibrosis is not discussed.

Response:   

Thanks for your comment. The third paragraph of the discussion has been amended to provide a more detailed account of the issue. “Many immune cells engage in constant communication with the gut microbiota within the gastrointestinal tract. The maturation of the immune system needs the development of commensal microorganisms. Furthermore, the gut microbiota is capable of mediating neutrophil migration and influencing T-cell differentiation, which may induce an immune response and stimulate inflammation or chronic tissue damage [4]. The skin microbiota is likely to affect many immune-related properties of epithelial health that are not yet appreciated [5]. Dysbiosis can disrupt cutaneous homeostasis, leading to leaky epithelium. It enables the penetration of microorganisms or their metabolites through the intercellular cracks in the skin, resulting in crosstalk between the microbiota and the immune system [6,7]. According to mediation analysis, the myeloid cells subtype could mediate the impact of skin microbiota on skin fibrosis. It has been reported that the infiltration and maturation of myeloid cells play a role in fibrotic repair [8]. Myeloid cells are a broad category of immune cells derived from myeloid progenitors in the bone marrow, including neutrophils. Myeloid cells function as potent producers of pro-inflammatory or anti-inflammatory factors, actively contributing to the pathogenesis of inflammatory diseases [9]. Neutrophils play a pivotal role in tissue restoration during wound healing by limiting microbial invasion at sites of skin or mucosal injury [10]. This offers a novel avenue for research into the treatment of skin fibrosis and the postponement of senescence. Furthermore, a more profound comprehension of the interactions between the microbiota and immune cells may facilitate the development of more efficacious treatments.

Q4.  How are the biomarkers above involved in mediating skin fibrosis (skin scarring), which is pathological wound healing, where TGF-beta is released from macrophages that involve AKT/mTOR and SMAD pathways?

Response:   

Thanks for your professional comment. The TGF-beta released from macrophages is the key to skin fibrosis. However, the samples of immune cells traits were collected from peripheral blood and analyzed by flow cytometry [11]. The present study is unable to provide direct evidence for the process in question.

Many immune factors play a role in the process of wound healing, such as IL-1β and IL-10. IL-1β can trigger the proinflammatory macrophage phenotype leading to aberrant wound healing [12]. IL-10 can improve wound healing or induce fibrosis by stimulating M2 macrophage polarization [13,14]. S. aureus α-toxin can induce IL-1β production from monocytes. By contrast, when exposed to S. aureus-derived cell wall component lipoteichoic acid, T cells neither proliferated nor produced cytokines. The gamma-proteobacteria genus Acinetobacter could induce anti-inflammatory immune responses that were protective against allergic inflammation. Besides, the presence of gamma-proteobacteria in the skin are correlated with greater IL-10 expression in blood [3,15]. Probiotics supplementation can modulate immune system towards anti-inflammatory response and synthesize beneficial anti-inflammatory metabolites. Moreover, it can also decrease cutaneous arterial sympathetic nerve tone and, increased cutaneous blood flow and skin hydration as well as TGF-β levels [16]. These showed that the microbiota was involved in the pathogenesis of dermatological conditions, with immune system acting as mediators. However, the mechanism between skin health and immunological responses caused by the skin microbiome is still largely unclear and requires further research [17].

Q5. Conclusions:

This section is vague.  While this topic is important to study, what is the value to the readership in how this applies to skin fibrosis now and in the future?

Response:

We appreciate your valuable comment. The section has been rewritten.

As the origin of skin diseases, a leaky epithelium is the new source of inspiration for explaining and fighting skin fibrosis and senescence. Microbial dysbiosis can disrupt cutaneous homeostasis leading to leaky epithelium.

There is a casual association between skin microbiota and skin fibrosis with immune cells acting as mediators. Skin microbiota has a pivotal function in skin fibrosis and can induce an immune response in the process. It is crucial to investigate the role of specific skin microbiota in dermatological conditions for precise treatment of skin fibrosis and the prevention of skin senescence.

For the suspected population, the skin microbiota can be collected by skin swabs for early screening and diagnosis. For patients, we propose novel and effective therapeutic strategies targeting specific skin microbiota. The effects of treatment can be evaluated by detecting the relative abundances of skin microbiota.

In future research, single-cell technologies combined with genomic, transcriptomic, proteomic and metabolomic analyses will clarify the crosstalk between the microbiota and the immune system, contributing to exploring the mechanism of skin fibrosis. It can provide insights into a novel target for skin senescence.

Above all, we appreciate your kindly and professional review from the bottom of our hearts. Thank you very much for your help.

Reviewer 2 Report

Comments and Suggestions for Authors

1.      The introduction should more explicitly describe the relationship between the microbiota and immune response. For improved clarity and to provide a concrete example, please include specific details or findings that illustrate how variations in the microbiota might influence immune responses.

2.      To enhance the comprehensibility of the study, a flowchart outlining the experimental design and data analysis steps is recommended.

3.      The conclusion should provide a detailed summary of the key findings of the study and explicitly articulate their significance in the context of the field. Additionally, it would be beneficial to include the implications of these results for future research and practical applications.

Comments on the Quality of English Language

Minor editing of English language required.

Author Response

Dear Reviewer:

Many thanks to the peer expert for reading our manuscript and reviewing it, which will help us improve it to a better scientific level. We revised our manuscript, and quite a lot of changes have taken place. We have resubmitted the revised manuscript and changes to our manuscript are all highlighted within the document by using red coloured text. We hope that the new manuscript will be acceptable to you. The following, points mentioned by the reviewer will be discussed:

Q1. The introduction should more explicitly describe the relationship between the microbiota and immune response. For improved clarity and to provide a concrete example, please include specific details or findings that illustrate how variations in the microbiota might influence immune responses.

Response:

Thank you for your professional comment. We rewrote the third paragraph of the introduction to describe the relationship between the microbiota and immune response. “The immune system plays a pivotal role in regulating the host's interaction with the gut microbiota [1]. Microbiota was reported to interact with immune responses by promoting macromolecules and antigens through the epithelium. The flagellin of microbiota is recognised by TLR5 on B cells, which differentiate into cells capable of producing IgA, neutralising pathogens and preventing infection [2,3]. S. aureus α-toxin can induce IL-1β production from monocytes, activating the immune system. [4]. Moreover, SASP is an inflammatory mediator that can induce immune responses [5,6]. Microbial dysbiosis would promote SASP damage [7]. Microbiota can release proinflammatory microbial products into the bloodstream via the leaky epithelium, resulting in a cross-talking immune system [8]. Therefore, it is valuable to investigate the role of immune cells in the impact of skin microbiota on skin fibrosis.

Q2. To enhance the comprehensibility of the study, a flowchart outlining the experimental design and data analysis steps is recommended.

Response:

Thanks for your professional suggestion. A flowchart was added to outline the experimental design and data analysis steps in Figure 1A.

Q3. The conclusion should provide a detailed summary of the key findings of the study and explicitly articulate their significance in the context of the field. Additionally, it would be beneficial to include the implications of these results for future research and practical applications.

Response:  

Thanks for your valuable suggestion. We rewrote the conclusion.

As the origin of skin diseases, a leaky epithelium is the new source of inspiration for explaining and fighting skin fibrosis and senescence. Microbial dysbiosis can disrupt cutaneous homeostasis leading to leaky epithelium.

There is a casual association between skin microbiota and skin fibrosis with immune cells acting as mediators. Skin microbiota has a pivotal function in skin fibrosis and can induce an immune response in the process. It is crucial to investigate the role of specific skin microbiota in dermatological conditions for precise treatment of skin fibrosis and the prevention of skin senescence.

For the suspected population, the skin microbiota can be collected by skin swabs for early screening and diagnosis. For patients, we propose novel and effective therapeutic strategies targeting specific skin microbiota. The effects of treatment can be evaluated by detecting the relative abundances of skin microbiota.

In future research, single-cell technologies combined with genomic, transcriptomic, proteomic and metabolomic analyses will clarify the crosstalk between the microbiota and the immune system, contributing to exploring the mechanism of skin fibrosis. It can provide insights into a novel target for skin senescence.

Above all, we appreciate your kindly and professional review from the bottom of our hearts. Thank you very much for your help.

Round 2

Reviewer 1 Report

Comments and Suggestions for Authors

The authors have addressed all of the items in the evaluation

Author Response

Thanks for your help. We appreciate your kindly and professional review from the bottom of our hearts.